# Electronic maternal and child health application usability, feasibility and acceptability among healthcare providers in Amhara region, Ethiopia

**Esubalew Alemneh**[1], **Tegegn Kebebaw**[1], **Dabere Nigatu**[2]*, **Muluken Azage**[2], **Eyaya Misgan**[3], **Enyew Abate**[3]

**1** ICT4D Research Center, Bahir Dar Institute of Technology, Bahir Dar University, Bahir Dar, Ethiopia, **2** School of Public Health, College of Medicine and Health Sciences, Bahir Dar University, Bahir Dar, Ethiopia, **3** School of Medicine, College of Medicine and Health Sciences, Bahir Dar University, Bahir Dar, Ethiopia

* daberen@yahoo.com

## Abstract

An innovative electronic Maternal and Child Health (eMCH) application was developed to support operational and clinical decision-making in maternal and child health services. End-user-based evaluation of eHealth application is a critical step to ascertain how successfully users can learn and use it, and improve the technology. Therefore, this study aimed to evaluate the eMCH tool usability, feasibility, and acceptability among healthcare providers (HCPs) in the Amhara region, Ethiopia. A cross-sectional study was conducted among HCPs working in six public healthcare facilities. The usability evaluation was done on 24 HCPs across three professional categories using the ISO 9241–11 usability guideline. One hundred nine HCPs were participated in the feasibility and acceptability study. Data were collected using a standard usability tool, think-aloud protocol, a self-administered approach, and Open Broadcaster Software Studio version 26.1.1 video recorder. Descriptive statistics were used to describe the data. A Kruskal-Wallis test was used to measure the association between mean scores and categories of HCPs. The recorded videos were used for the log file analysis method. None of the HCP categories were able to complete all the tasks without errors. The average number of errors and restarts were 7.5 and 2.8, respectively. The average number of restarts was directly proportional to the average number of errors. The participants successfully completed more than 70% of the tasks without requiring any assistance or guidance. Forty-seven comments or errors were identified from the think-aloud analysis and 22 comments from the usability metrics analysis. Overall, statistically significant performance differences were observed among the three HCP groups across the majority of the usability evaluation metrics. Fifty-seven percent of HCPs scored higher than the mean on the feasibility study. Slightly higher than half, 56 (51.4%), of the HCPs scored higher than the mean score on the acceptability study. The usability evaluation identified vital comments and usability flaws that were essential for the eMCH tool to be upgraded. The tool was feasible and acceptable as reported by end-users. Therefore, the errors and usability flaws of the tool should be fixed before deployment to other healthcare settings.

**Data Availability Statement:** All data are in the supporting information and manuscript.

**Funding:** TK received the research fund from the Bahir Dar University ICT4D research center. The funder has no role in the design, data collection, analysis, decision to publish, or preparation of the manuscript.

**Competing interests:** The authors have declared that no competing interests exist.

## Author summary

The study team developed an electronic Maternal and Child Health (eMCH) application to be used by healthcare providers (HCPs) serving mothers and their babies. The participation of end-users in the design and evaluation of a new application is a vital step to produce contextual, acceptable and easy to use application. Here, the study wants to evaluate the usability, feasibility, and acceptability of the eMCH application by healthcare providers. Twelve tasks were given to twenty-four healthcare providers who were selected from three provider categories. The HCPs had to use the eMCH application to complete the tasks. The study revealed that none of the HCP categories were able to complete all the tasks without any mistake. The number of eMCH application restarts was directly proportional to the number of errors committed. The study also identified many vital comments and usability flaws that were essential to upgrade the eMCH application. The results gave an insight that the comments and usability flaws of the application should be fixed before deployment to other healthcare institutions.

## Introduction

Digital health, which comprises emerging technologies and eHealth (such as mHealth, telemonitoring, video-consultations, electronic health records), play a crucial role to advance the core principle of primary health care (PHC)–a people-centered and integrated health service delivery model [1–3]. Many low-and middle-income countries (LMICs) are leveraging advances in digital health technologies to improve and maintain the continuity of service delivery post-COVID-19, by innovating, testing, evaluating, and, in some instances, integrating digital health solutions into PHC settings [4,5].

The use of digital health is also recognized and recommended as vital intervention to guide and support clinical and operational decision-making across the health system to reduce inequality and increase universal health coverage [6]. Digital health may improve healthcare quality in different areas of interventions such as patient safety, access to healthcare, effective treatment, efficient use of resources, equity of care across subgroups of populations, and patient-centered care [6–8]. The rise and use of digital health adoption may also have created a challenge to establish the extent of the impact of digital health on quality healthcare [5,8]. The challenges associated with digital health include ethical concerns about information security, the acceptability of new digital tools and practices, and whether the technology really is practical and feasible to be implemented in real practice [9]. End-users (i.e., healthcare providers) perspective is one component in the design and deployment of eHealth applications in healthcare facilities [10].

The study team developed an innovative eMCH (electronic Maternal and Child Health) which is an integrative, interoperable and vendor-neutral digital health platform to support operational and clinical decision-making in maternal and child health (MCH) services. The tool transformed the current paper-based information management system and services provision to digital system. As the eMCH tool stand now, the primary end-users are healthcare providers (HCPs) who are deployed in MCH unit. In eHealth development process, measuring usability, acceptability, and feasibility of the new tool among end-users is a vital step to determine how successfully users can learn and utilize the tool to accomplish their goals and to identify chances to improve the technology. Therefore, this study aimed to evaluate eMCH application usability, acceptability, and feasibility among HCPs in Amhara region, Ethiopia.

Addressing the users' comments and usability flaws will help to produce usable and acceptable eMCH tool that eventually helps to improve the quality of care and reduce mortality and morbidity of mothers and children.

## Methods and materials

### Study design and setting

A cross-sectional study was conducted from February 24, 2022 to March 08, 2022. The study involved healthcare providers working in 6 public health facilities (3 hospitals and 3 health centers) in Amhara region, Ethiopia. HCPs who were working in urban, peri-urban and rural setting healthcare facilities were participated in the study. The healthcare facilities were selected from Bahir Dar city, Bahir Dar Zuria district and North Mecha district. Healthcare providers who were working in MCH unit were participated in the study. The professional categories participated were midwives, integrated emergence surgery officers (IESOs) and medical doctors.

### Sample size and sampling procedure

All HCPs who were working in MCH unit in six public health facilities were considered in the feasibility and acceptability study. In the case of usability evaluation, representatives of real users were selected to conduct user-based usability testing. Scholarly studies suggested different number of users for usability testing [11–14]. Up to 80% of usability problems can be detected by involving 4 to 5 participants and the benefit/cost ratio declines as more participants are added [12]. Scholars in the field recommended that 15 to 20 study participants are adequate to detect all usability problems in user-based usability testing [11,13]. Splitting the usability testers into different groups is also found to be advantageous in usability testing [15]. Thus, 24 evaluators from the three categories of health professionals were involved in the usability evaluation. Initial screening questionnaire was used to identify and administer usability evaluation. The usability evaluation was done by 24 study participants who were fit for usability testing process.

### Digital health tool description

The eMCH application development process was started in 2020 by a team of experts. The team composed of experts from ICT, clinicians and public health professionals. The eMCH application, a quality improvement tool, was designed to be used at a point-of-care by healthcare providers. The tool had a decision support system to enhance healthcare providers' clinical decision. The eMCH tool included maternal continuum of care components: antenatal care (ANC), delivery care including ePartograph (electronic partograph), and postnatal care (PNC). The tool transformed the paper-based standard care into electronic form to guide and support maternal and newborn healthcare services. The detail description of ePartograph tool was presented in Kebebaw et al. 2021 [16]. A responsive web design approach was followed to develop the system. So that, the applications can run in devices of different screen sizes. The usability problems discovered in heuristic-based usability evaluation were corrected before the current end-user testing. A snapshot of filled ePartograph was given below (Fig 1). Additional interfaces from the application were also provided as supporting information files (see S1 Fig, S2 Fig and S3 Fig).

### Data collection tools and procedure

A self-administered questionnaire with five-point Likert scale was used to measure feasibility and acceptability of eMCH application by healthcare providers. The scale was developed by

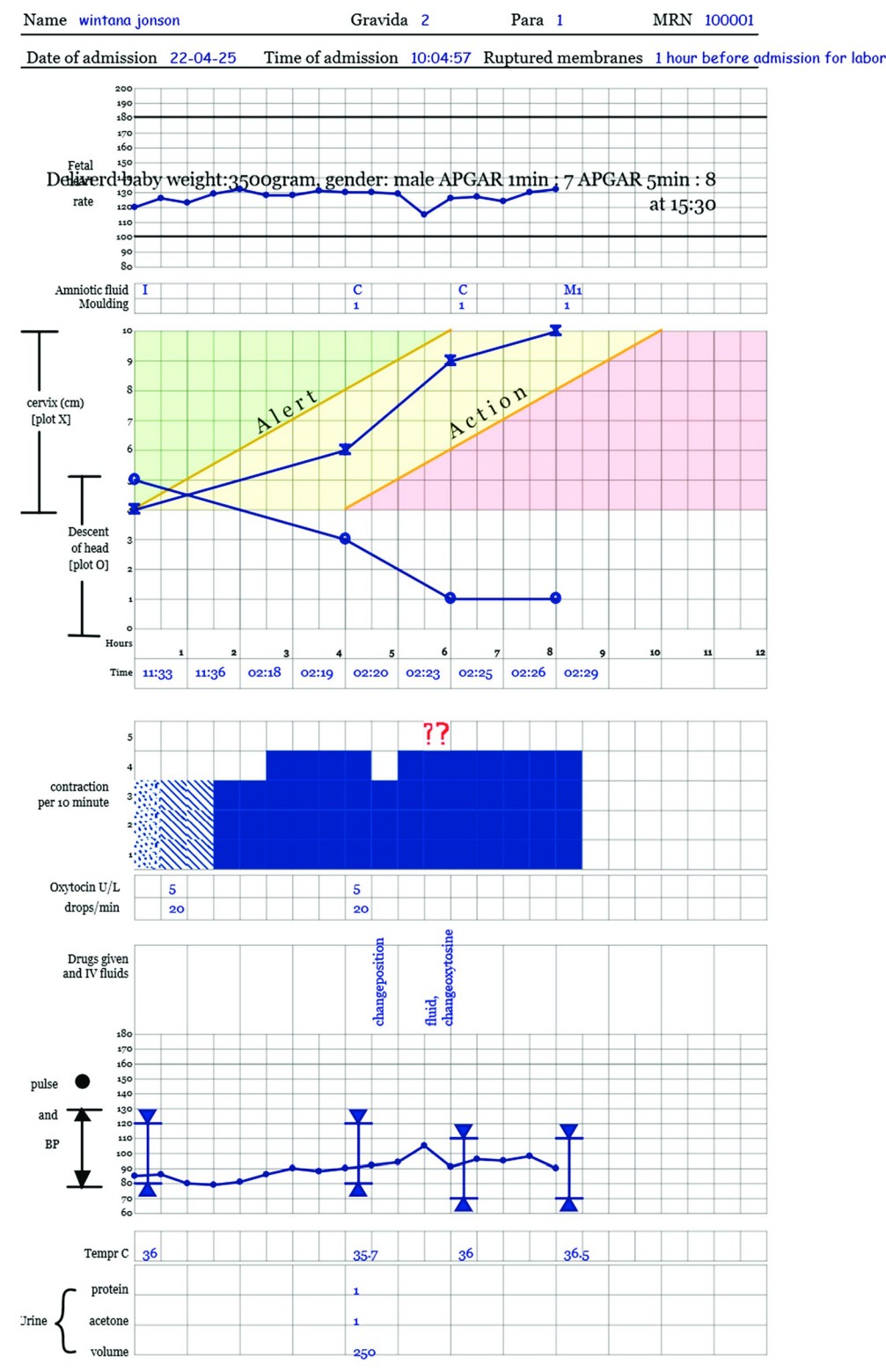

**Fig 1. A snapshot of filled eMCH (ePartograph).**

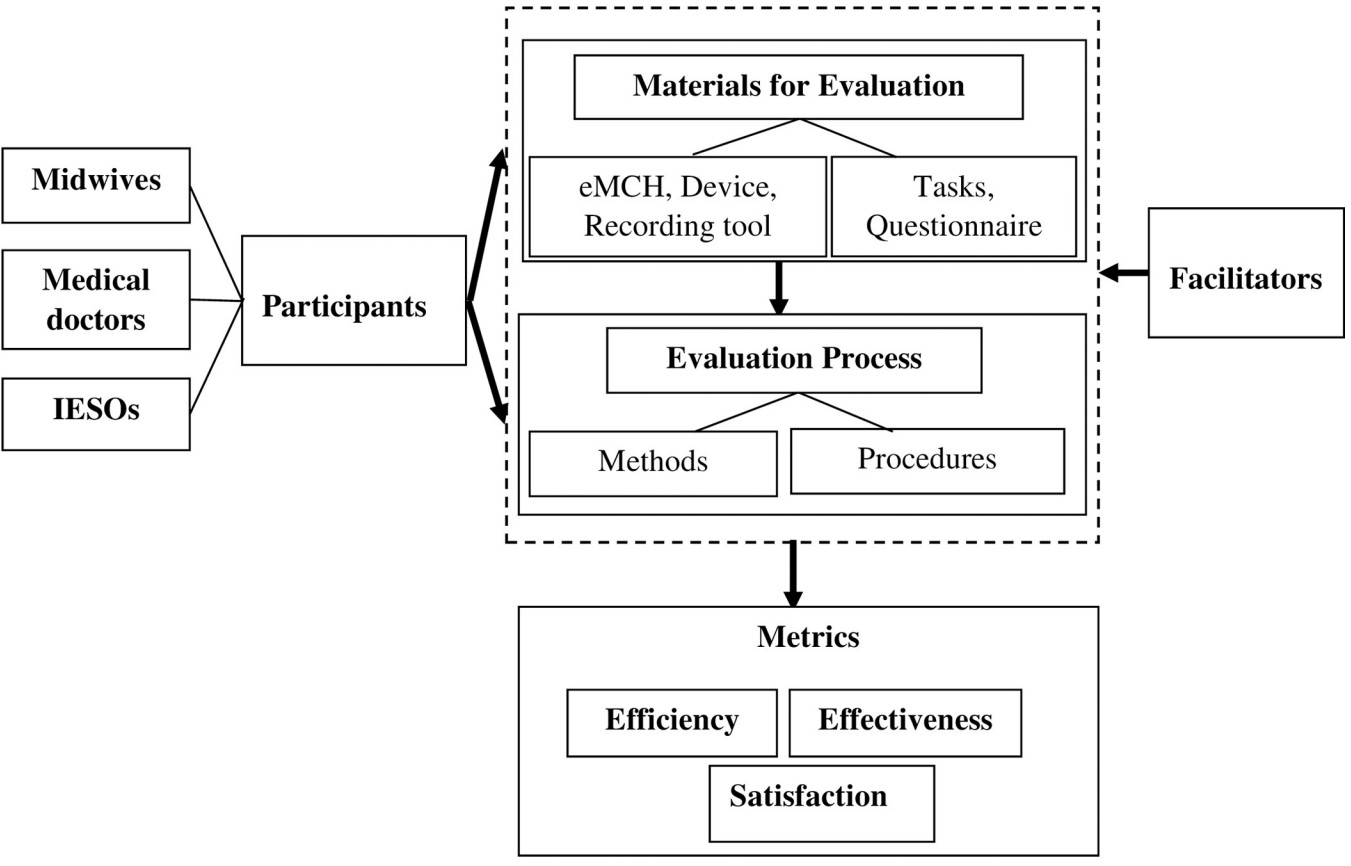

**Fig 2. Usability evaluation framework.** IESOs: Integrated emergency surgery officers; eMCH: electronic Maternal and Child Health.

reviewing literature. The data were collected using a self-administered English version questionnaire. The measurement tool had 23 items: 10 items were designed to assess feasibility of the eMCH tool while 13 items were designed to assess acceptability of the eMCH application. Six investigator team members facilitated the data collection procedure.

In this usability evaluation, think-aloud protocol, observation, log analysis and questionnaire were applied to capture usability evaluation data. Usability evaluation framework shown on the diagram below was used to conduct the usability evaluation (Fig 2). The end user-based usability evaluation was assessed in terms of efficiency, effectiveness and satisfaction. Evaluation participants executed a set of tasks on the eMCH using thinking aloud approach. Facilitators observed, guided, and monitored the overall evaluation process.

The eMCH tool can run in devices of different screen sizes. For this study, however, 7-inch tablets with a screen size of 7.5" x 4.75" were used. In the think-aloud testing, also known as think-aloud protocol, the evaluators speak aloud about their thinking throughout the evaluation process. The think-aloud approach is the most widely used usability testing method [17]. It is known to give detail understanding of usability problems as it helps to identify actual difficulties end-users come upon and the causes of underlying problems. The think-aloud usability evaluation method stems from the field of cognitive psychology and it is flexible, cheap, easy to learn, and convincing usability evaluation method [18,19]. This day, the think-aloud protocol is becoming a well-recognized usability evaluation method to capture data on user experiences of electronic applications [20–23]. Open Broadcaster Software (OBS) Studio version 26.1.1 was used to record the end-users' responses and their interactions to complete the tasks. The

software was launched from the home screen and the recording mode was selected from virtual camera screen. When a user was ready, the facilitators hit start recording button, and the software immediately hidden any tool interface and start to display the evaluator's desktop screen. The OBS tool recorded every action including voices.

Two sets of questionnaires were used to gather information about participants and their experience of the system. The first questionnaire contains open ended questions that were used to gather demographic information and computer skills and usage. The second questionnaire contains the standard task level satisfaction questions (i.e., Single Ease Question (SEQ)) [24] and test level satisfaction questions(i.e., System Usability Scale (SUS)) [25].

The usability evaluation was conducted in two sessions in the presence of four domain experts and two usability experts as a facilitator. Midwives and general practitioners form separate sessions and IESOs were evenly distributed into the two groups. For each session two domain experts and one usability experts were assigned. The facilitators delivered a 30 minutes orientation for usability evaluators. The orientation included explanation of the application, the tasks scenarios, and the usability testing process. The evaluation process was started following the orientation. As first step the participants started to fill demographic information as well as computer skills and usage information. Next, they went through the task scenario and executed them while uttering their views as they move through the user interface. The software captures every interaction of the evaluators with the system. The OBS recorder was re-launched either by the participant or with the help of facilitators when the participant suddenly closed the software.

During the testing session, mobile devices were silenced to avoid distraction. Generally, the two evaluation sessions were conducted in a controlled environment. Twelve tasks which were supposed to be carried out using the eMCH were given for participants. The tasks were proposed based on relevance to MCH care professionals' daily practice. The tasks were categorized into two roles with detail scenario descriptions. Some of the task scenarios are portrayed in table (Table 1). The users filled the post-test questions after completing the tasks. The usability evaluation questionnaires used were valid and reliable instruments to measure the satisfaction of users [26].

## Study variables and metrics

In this study, usability was measured following the ISO 9241–11 usability framework (Fig 3). Usability metrics involve measurement of usability attributes like effectiveness, efficiency, and satisfaction of users of a product [27]. Efficiency was measured in terms of time taken to

**Table 1. Tasks for usability evaluation of eMCH application by healthcare providers in Amhara region, Ethiopia.**

| Task ID | Task | Scenario | Role |
|---|---|---|---|
| T1 | Register a patient | Register a woman with the following details: Name = "Maeza Debebe Zeru", Date of Birth ="23/09/1991", Phone Number ="+251-911-901055", Gravida = "3", Para = "2", Ruptured Membrane ="No, intact", Next of Keen = "Debebe Zeru", Keen's Phone Number = "+251-911-901056" | Recorder |
| T3 | Assign/Revoke HCP for patients | Assign a caregiver named "Zewdu Ayel" to a patient with MRN number "11009". Change care provider of the same patient from "Zewdu Ayel" to "Addisu Damena" | Recorder |
| T7 | Record first time 4-hour examination data | Make the 4-hour registration for a patient assigned to you. Cervix Dilation ="4", Decent of Head ="4" Amniotic Fluid =" I", Molding ="1", Contractions per 10 minutes: $1^{st}$ ="<20"; 2nd ="20–40";3rd ="20–40", Heart Rate ="90", Fatal Heart Rate ="154", BP [Systolic/Diastolic] ="113/79", temperature ="37", Urine: protein ="200"; acetone ="20", Volume ="51" | Provider |
| T10 | View patient information in partograph | Search for a patient with MRN number 1101 and view her partograph | Provider |

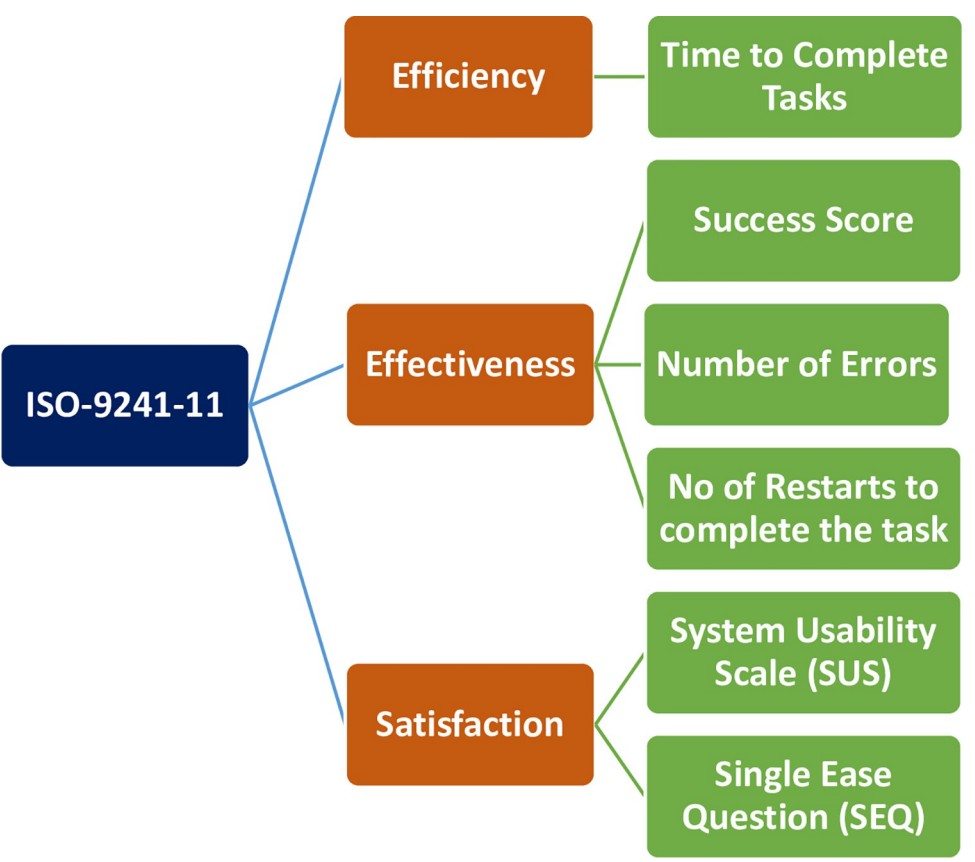

**Fig 3. Usability metrics used in the study.**

complete the tasks. Effectiveness was measured in terms of task completion rate (success score), number of errors an evaluator committed and number of restarts to complete a task. The study used the two most commonly used satisfaction assessment scales: SUS and SEQ. The SUS is 5-point Likert scale which comprises of 10 questions that are used to assess test level usability and learnability of a product. The SEQ on the other hand is a 7-point Likert scale and it is used to measure task level satisfaction.

The four-usability metrics were defined as follows:

- *Number of errors*: the number of errors a user has committed while trying to accomplish a task was counted. Some examples of the errors were clicking wrong link or unclickable image, clicking save/complete and other buttons before entering/selecting all mandatory fields, entering/selecting wrong values, and double clicks.

- *Task completion time*: it was calculated as a difference between task completion time and task start time. Start time was counted from the moment the evaluator has finished reading task scenario and completion time was the time user has declared she/he has finished the task verbally.

- *Success score*: this metrics measures average successful task completion rate per a user. Tasks were completed with no facilitator intervention, with little assistance or with guidance (giving detailed directions for user on how to complete the tasks).

- *Number of restarts to complete the task*: this metrics refers the number of restarts made to complete the task.

In this study, feasibility was measured by computing 10 questions with five-point Likert scale. The participants rated each question 1 to 5: very easy (5), easy (4), undecided (3), difficult (2), very difficult (1). The higher the score the more feasible was the eMCH application. The eMCH application was considered feasible if a participant scores greater or equal the mean value. Similarly, the acceptability was measured by computing 13 questions with five-point Likert scale. The participants rated each question 1 to 5: strongly agree (5), agree (4), undecided (3), disagree (2), strongly disagree (1). The higher the score the more acceptable was the eMCH application. The eMCH application was considered acceptable if a participant scores greater or equal the mean value.

## Data management and analysis

The feasibility and acceptability data were coded and enter to SPSS version 21. Descriptive statistics, such as mean, frequencies, percent, were used to analyze feasibility and acceptability data. Bar graphs and tables were used to present the data. Additionally, Kruskal-Wallis test (a non-parametric test) [28] was used to measure the association between mean scores of errors and satisfaction, and professional categories.

The log file analysis method was used to analyze the recorded video. The metrics' values were identified by log file analysis, and usability problems were jotted down. The recorded videos were carefully and independently evaluated by two experts and the values were extracted. Whenever the video reviewers detect discrepancies of the values, the videos were rechecked for correct values. For efficiency and effectiveness, the analysis was made for each task. The study took either the average or count of the values based on the type of metrics used. Tasks involving role switching (recorder to provider or vice versa) was ignored from the analysis.

## Ethics statement

Ethical approval was obtained from Bahir Dar University College of Medicine and Health Sciences Institutional Review Board. Support letters were obtained from Amhara Public Health Institute. Then, a permission letter was sought from all relevant healthcare facilities. Besides, written informed consent was obtained from each study participant. Anonymity and confidentiality of information was ensured throughout the study process.

## Results

### Sociodemographic characteristics of study participants

The feasibility and acceptability questionnaire were completed by 109 healthcare providers. Forty-five (41.3%) of the healthcare providers were in the age range of 30 to 34 years old. Seventy (64.2%) were females, 100 (91.7%) were first degree holders and 90 (82.6%) were midwives by profession (Table 2).

Twenty-four healthcare providers were involved in the usability evaluation. Of the 24 HCPs, 12 were females and 13 were first degree holders. Majority, 16 HCPs had intermediate computer skill while 9 healthcare providers had 2 to 3 hours exposure to computer use per day (Table 3).

### Feasibility and acceptability of eMCH

The mean feasibility score was 38.7 (standard deviation (SD) = ±5.8). The study revealed that 62 (56.9%) healthcare provider scored higher than the mean. The mean acceptability score was

**Table 2. Sociodemographic characteristics of healthcare providers involved in feasibility and acceptability study, Amhara region, Ethiopia.**

| Variables | Frequency | Percentage |
|---|---|---|
| **Age of respondents** | | |
| 20–24 | 4 | 3.7 |
| 25–29 | 39 | 35.8 |
| 30–34 | 45 | 41.3 |
| 35–39 | 12 | 11.0 |
| > = 40 | 9 | 8.2 |
| **Sex of respondents** | | |
| Female | 70 | 64.2 |
| Male | 39 | 35.8 |
| **Qualification of respondents** | | |
| First degree | 100 | 91.7 |
| Second degree and above | 9 | 8.3 |
| **Professional category** | | |
| Midwives | 90 | 82.6 |
| IESOs | 9 | 8.3 |
| GPs | 10 | 9.1 |

IESOs: Integrate emergency surgery officers; GPs: General practitioners

**Table 3. Demographic characteristic of usability study participants by professional category, Amhara region, Ethiopia.**

| Variables | Midwives | | IESOs | | GPs | |
|---|---|---|---|---|---|---|
| | Number | % | Number | % | Number | % |
| **Sex** | | | | | | |
| Male | 2 | 25 | 5 | 62.5 | 5 | 62.5 |
| Female | 6 | 75 | 3 | 37.5 | 3 | 32.5 |
| **Age (in years)** | | | | | | |
| 20–30 | 4 | 50 | 2 | 25 | 3 | 57.5 |
| 31–45 | 3 | 37.5 | 4 | 50 | 5 | 62.5 |
| > 45 | 1 | 12.5 | 2 | 25 | - | - |
| **Level of Education** | | | | | | |
| BSc | 5 | 62.5 | - | - | - | - |
| MSc | 3 | 37.5 | 8 | 100 | - | - |
| MD | - | - | - | - | 8 | 100 |
| **Computer usage per day** | | | | | | |
| < 1 hours | 2 | 25 | 1 | 12.5 | 1 | 12.5 |
| 1–2 hours | 3 | 37.5 | 3 | 37.5 | 3 | 37.5 |
| 2–3 hours | 3 | 37.5 | 3 | 37.5 | 3 | 37.5 |
| >3 hours | - | - | 1 | 12.5 | 1 | 12.5 |
| **Computer Skills** | | | | | | |
| Basic | 2 | 25 | 1 | 12.5 | - | - |
| Intermediate | 6 | 75 | 5 | 62.5 | 5 | 62.5 |
| Advanced | - | - | 2 | 25 | 3 | 37.5 |
| **Paper partographs filled per day** | | | | | | |
| < 5 | - | - | 3 | 37.5 | 3 | 37.5 |
| 5–10 | 3 | 37.5 | 4 | 50 | 3 | 37.5 |
| 10–15 | 5 | 62.5 | 1 | 12.5 | 2 | 25 |

IESOs: Integrate emergency surgery officers; MD: Medical doctor; GPs: General practitioners

**Table 4. Feasibility and acceptability of eMCH by healthcare providers in public health facilities, Amhara region, Ethiopia.**

| Variables | Frequency | Per cent |
|---|---|---|
| **Feasibility score** | | |
| Below mean | 47 | 43.1 |
| Mean and above | 62 | 56.9 |
| **Acceptability score** | | |
| Below mean | 53 | 48.6 |
| Mean and above | 56 | 51.4 |

52.8 (SD = ±6.9). Slightly higher than half, 56 (51.4%), of the healthcare providers scored higher than the mean score (Table 4).

## Usability evaluation

The average number of errors and restarts were 7.5 and 2.8, respectively. The IESOs made more slips or errors in comparison to the other two professional groups. The Kruskal-Wallis test on both metrics showed a statistically significant difference in number of errors committed ($X^2$ = 7.86, P = 0.019) and number of restarts between professional groups ($X^2$ = 11.00, P = 0.004) (Table 5).

The study revealed that none of the professional categories were able to complete all the tasks without assistance or guidance. Some required guidance to complete the tasks. On the other hand, all the professional groups were able to complete, on average, more than 70% of the tasks without requiring any assistance or guidance. Medical doctors outperform other groups in successful completion of tasks (81%). Midwives tend to complete their tasks after some support or assistance (18%, 13 out of 72 instances). Medical doctors were the groups that required minimum guidance (4%, 3 out of 72 instances). The IESOs on the other hand required the highest guidance to complete the task. More specifically at least one participant required guidance to accomplish most of the tasks. Task completion rate significantly varied among professional groups ($X^2$ = 6.78, P = 0.0337) (Fig 4).

The result revealed that midwives generally tend to take more time (average completion time 81.5 seconds) to complete most of the tasks. Medical doctors were the fastest group (average completion time 53.5 seconds) to complete majority of the tasks. Task 1 and Task 7 take higher time because the two tasks consist steps that involve filling forms. The task completion time varied among professional categories ($X^2$ = 6.48, P = 0.039). The mean task completion time among midwives, IESOs, and GPs were 24.3, 17.5 and 13.5 respectively. The pair-wise completion showed the existence of significant difference between midwives and GPs (p = 0.006) (Fig 5).

**Table 5. Average number of errors and restarts per professional categories, Amhara region, Ethiopia.**

| Variables | Average number of errors | $X^2$ (P-value) | Average number of restarts | $X^2$ (P-value) |
|---|---|---|---|---|
| **Professional category** | | | | |
| Midwives | 7.63 | 7.86 (0.019) | 2.38 | 11.0 (0.004) |
| IESOs | 8.75 | | 3.25 | |
| GPs | 6.25 | | 1.5 | |
| **Average score** | **7.54** | | **2.38** | |

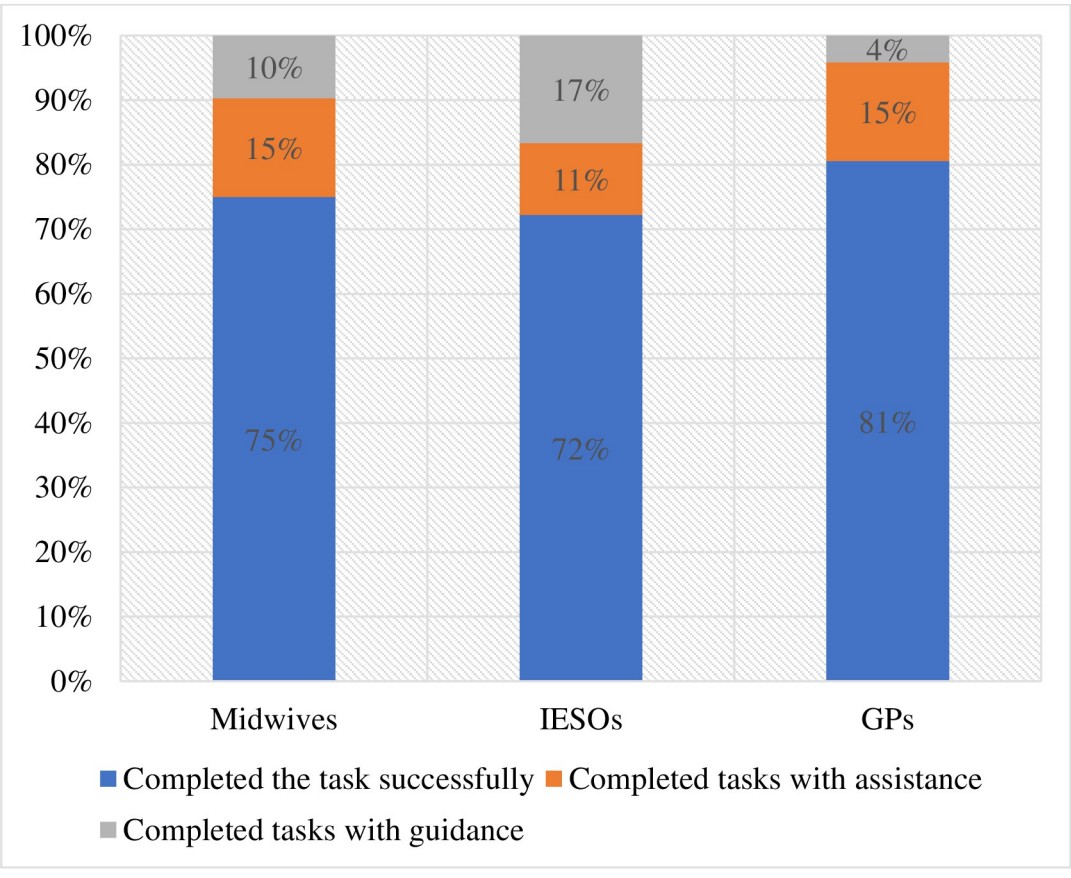

**Fig 4. Task completion rate of healthcare providers by their profession, Amhara region, Ethiopia.** Guidance refers when eMCH user required extensive guide or action from facilitators to accomplish a task given whereas assistance refers when eMCH user required a minimal help from facilitators to accomplish a task given. IESOs: Integrate emergency surgery officers; GPs: General practitioners.

The result showed that midwives and medical doctors found the tasks easier to use than the IESOs. There was no statistically significant difference between professional categories ($X^2 = 5.52$, P = 0.063) (Fig 6).

Medical doctors scored the highest satisfaction in using the eMCH application while IESOs scored the lowest. The Kruskal-Wallis test depicted that there was statistically significant difference in satisfaction level among professional groups ($X^2 = 6.65$, P = 0.036) (Fig 7).

Additionally, 47 comments or errors from the think-aloud analysis and 22 comments from usability metrics analysis were identified. The nine most frequent comments or errors from the think-aloud recording are presented in table (Table 6). Majority, 63% of the evaluators want the system to be accessed via their local language, Amharic. Especially, 75% of the midwives prefer accessing the tool using Amharic. The other comment which was reiterated by 13/24 (54%) evaluators was the system shall restrict entering subsequent contractions before preceding ones were entered. Forty-six percent of the participants suggested one decimal place for temperature values. Forty-two percent of the participants want the tool display the clinical intervention provided for patients or clients (Table 6).

The study also identified different comments or errors from selected usability evaluation tasks with their suggested remedials to upgrade the tool. The top four tasks that scored highest

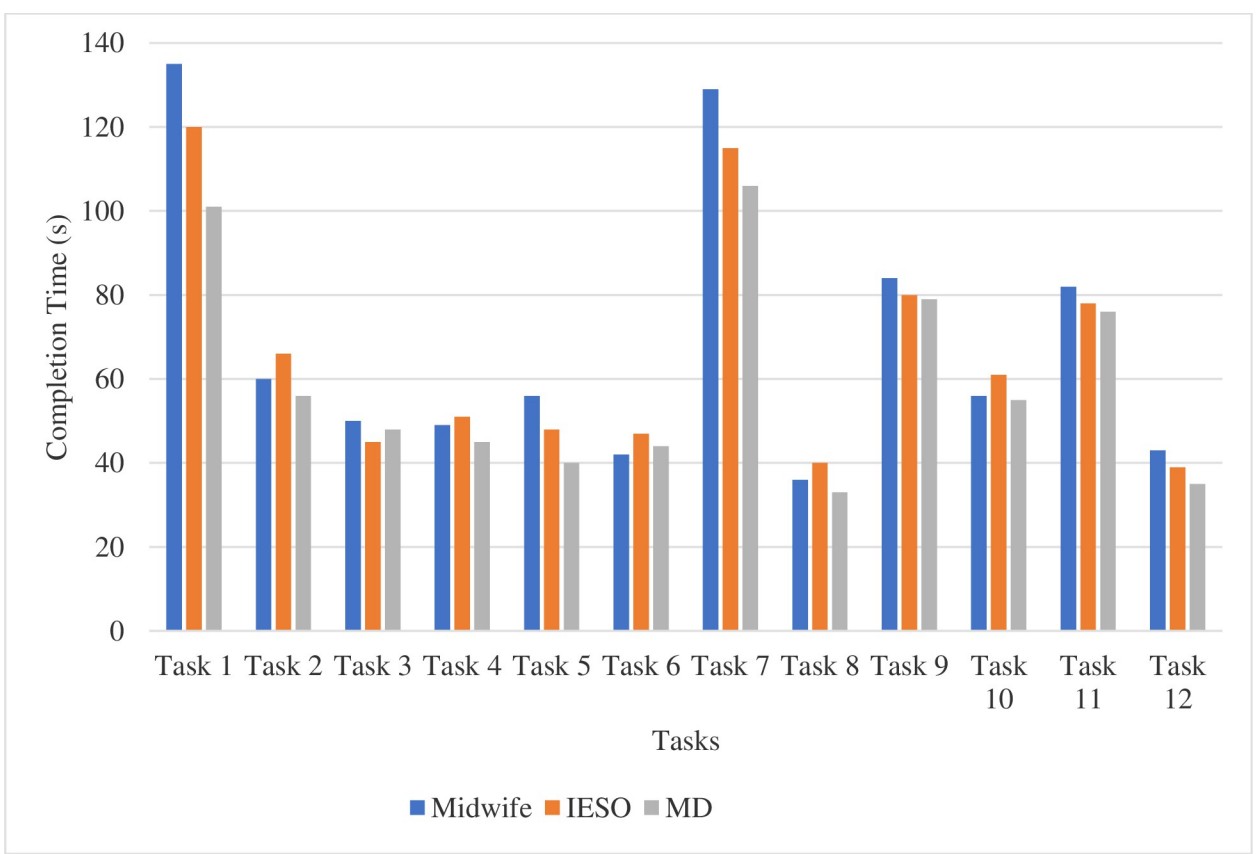

**Fig 5. Average task completion time by professional categories, Amhara region, Ethiopia.** IESOs: Integrate emergency surgery officer; MD: Medical doctor.

average completion time by the three groups of professionals were summarized in table below (Table 7).

## Discussion

The study revealed that none of the user groups were able to complete all the twelve experimental tasks without errors. The average number of errors committed was 7.54. The finding is consistent with the previous study which showed that the average number of errors per task was 0.7 [29]. The study also revealed that the average number of restarts were directly proportional to average number of errors. Those who have made more mistakes have restarted their tasks more frequently. This indicates that making slips and mistakes when performing tasks is perfectly normal. Though no severe error was detected, it helped to obtain minor errors that needs to be diagnosed. For example, most users made errors in entering date values. They either mixed Ethiopian Calendar with Gregorian Calendar or typed wrong date formats. This has given a lesson to explicitly mention the calendar type and to change the date field from textbox to date picker. The study showed that number of errors detected had statistically significant variation across the study groups. The variation could be related to practical partograph usage experience by study groups.

Task completion rate or success rate has strong correlations with usability benchmarks for eHealth systems [29]. This study revealed that two of the three groups failed to meet the threshold value of a usable application (78%) as suggested in Sauro 2010 [30]. This might be

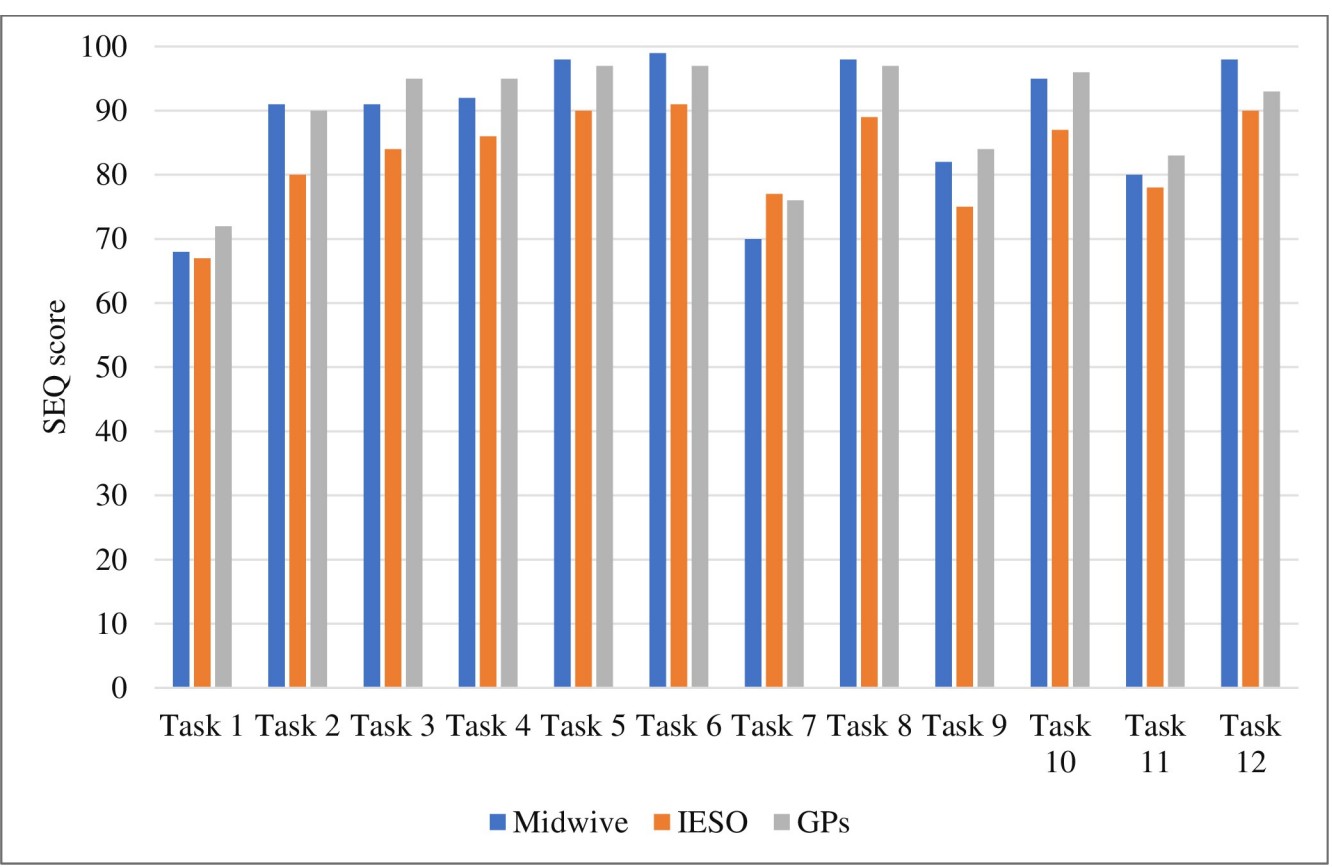

**Fig 6. Average ease of use of each task by professional categories, Amhara region, Ethiopia.** IESOs: Integrate emergency surgery officer; GPs: General practitioners.

due to unfamiliarity to the eMCH tool. The study assumes that after some training and experience they can use the tool effectively as no serious usability flaw was detected in the analysis of the recording of participants thinking-aloud.

The study showed that all professional groups involved were able to complete, on average, more than 70% of the tasks without requiring any assistance or guidance. Medical doctors had the highest performance in successful completion of tasks compared to other groups. The IESOs on the other hand required the highest guidance to complete the tasks. More specifically at least one participant required guidance to accomplish most of the tasks. It is obvious that task completion time depends on factors related to the type and complexity of task, the participant, the device used, and context of use. Bad user interface also usually leads to longer task completion time [31]. Longer task completion time may in turn reveal symptoms of user interface problems. However, the metrics doesn't state exactly where the problems are. Further investigation can be undertaken to detect the flaws on usability for eventual diagnostic measure in formative tests. For instance, adding autofill and exhaustive listing of values for some fields for sure will enable users accomplish their tasks faster for tasks like Task 1 and Task 2.

The study showed that satisfaction rating (SUS) varied by the study groups which is consistent with previous studies [32,33]. The study also did not find the correlation stated between SUS and task performance as did in Sauro 2009 [29]. Just like other usability metrics, post-test satisfaction metrics tells the overall satisfaction opinion about the application but it fails to provide particular symptomatic information. However, if the result is very low user experience

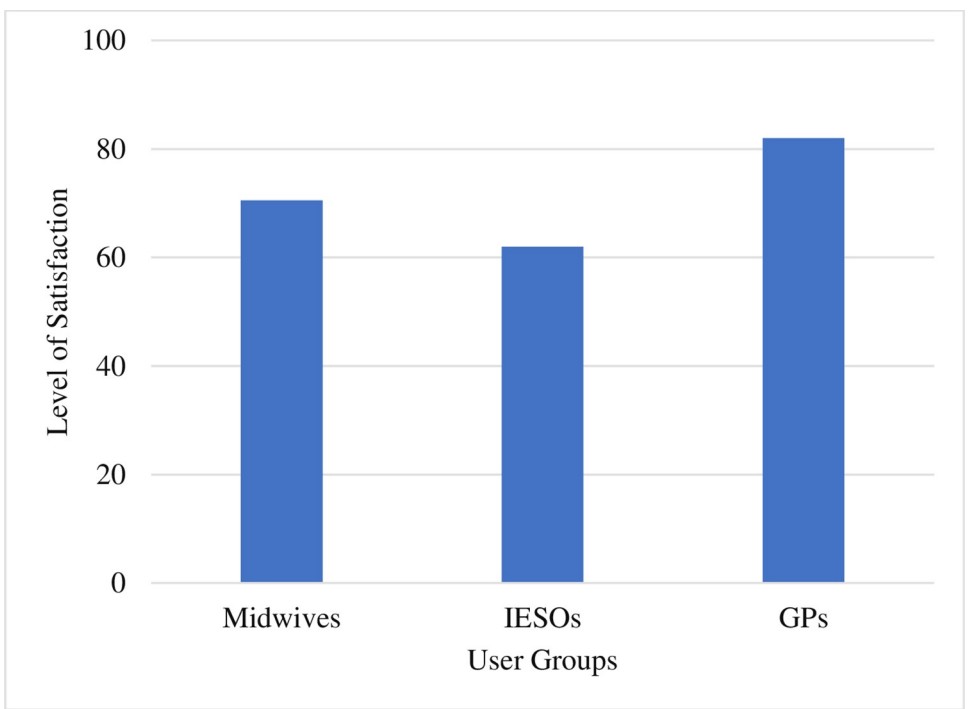

**Fig 7. The average SUS result by professional categories, Amhara region, Ethiopia.** IESOs: Integrate emergency surgery officers; GPs: General practitioners.

practitioners can make investigations on tasks that are completed with assistance and guidance to find out user interface problems. For example, they may check if labels are clear and comprehensible, or if the tasks are complex.

The results also depicted that though midwives were frequent users of partographs their performance didn't reflect their usage experience. This might be due to their computer skills and time spent on computers. On contrary, despite the fact that IESO have better computer skills and spend more time on computers, they were less performing group in all measures. This is more probably due to lack of practical experience on partograph usage. Medical doctors performed very well in most of the usability metrics. Listing usability issues from think-aloud protocols remains one of the most effective tools to explain the usability for eHealth [34]. The study identified various errors from selected usability evaluation tasks. The study also

**Table 6. Usability issues obtained by analyzing think-aloud recordings of healthcare providers, Amhara region, Ethiopia.**

| Comments/Recommendations | Midwives | IESOs | GPs | Sum | % |
|---|---|---|---|---|---|
| The system should also work in Amharic Language | 6 | 5 | 4 | 15 | 63% |
| It shall not allow filling contractions before preceding contraction is filled, i.e., unless contraction 2 filled the system shouldn't accept 3rd, 4th, or 5th contractions | 4 | 4 | 5 | 13 | 54% |
| One decimal place is sufficient to specify temperature | 4 | 4 | 3 | 11 | 46% |
| The visibility of the time left for 30-minute and 4-hour examination should be improved | 5 | 3 | 3 | 11 | 46% |
| No way to cancel error and warning messages | 5 | 4 | 2 | 11 | 46% |
| The intervention medicines provided to patients should be visible at the drug list | 3 | 3 | 4 | 10 | 42% |
| In assigning caregiver, the default caregiver in drop down should be "Select caregiver" than displaying selected caregiver name | 2 | 3 | 2 | 7 | 29% |
| Maternal pulse rate won't be filled every 30 minutes based on the current practice | 0 | 2 | 3 | 5 | 21% |
| Assigning a caregiver misses some information displayed on all pages | 1 | 0 | 2 | 3 | 13% |

**Table 7. Usability issues extracted from usability tasks performed by healthcare providers, Amhara region, Ethiopia.**

| Task | Comment/Error | Recommendation |
|------|---------------|----------------|
| **Task 1** | Mixing Ethiopian and Gregorian Calendars | Date types should be explicitly stated |
| | Mistakes in date formats | Dates should be picked instead of typing them out |
| | Typing first, first name, then middle name and finally family name takes time | First, middle and last names be typed in single field |
| | Many users miss or add one digit to the phone numbers | Format the phone numbers and validate the inputs |
| | Can't see registration date and time on registration page | Display registration date and time on the page instead of internally handling it |
| **Task 7** | Wrong values were entered for cervix and molding fields | Cervix dilation values should be populated with correct values between 4 and 10, inclusive. Molding is also to be changed to dropdown control with values 0, 1, 2 and 3 |
| | Some fields take invalid values | Add validation for all fields |
| | Sliding a control for heart rates and BP takes time | Change this fields to editable field |
| | Warning not shown for abnormal values on the that page | Modify the system in such a way that exaggerated values are displayed to warn caregivers |
| **Task 9** | As intervention dialogue is transparent, the text on the dialogue is not readability | Increase opacity of the dialogue box so that readability is not distracted by background texts |
| | The tool displays all possible interventions instead of suggesting the most appropriate ones based on the measured values | Recommendation of the best intervention based on examination values is beyond the scope of this system. |
| **Task 11** | Fields for "Contraction Per 10 minutes" are clustered even if there is large open space | Arrange 1st to 5th contractions vertically instead of arranging them in a row |
| | Time left for 30 minutes examination should be shown on partograph page | Partograph page should be modified to show time left for 30 minutes and 4-hour examinations |

uncovered the top four tasks that have highest average completion time by the three groups. They were activities that were found difficult and were most likely to be the main reasons for low success rates and high error rates which is consistent with the previous study [29]. The remedial suggestions also advised to upgrade the tool based on the comments received from usability evaluators.

The study has some limitations. Firstly, the eMCH application usability on small-sized device (smartphones) has to be studied since the current study was conducted on a 7-inch tablet with a screen size of 7.5" x 4.75". Usability evaluation on small-sized devices may facilitate implementation of eMCH tool by smartphones since almost all healthcare providers have smartphones and its potential to overcome the observed resources constraints in public healthcare facilities. Secondly, the sample size may not be adequate to use parametric tests.

## Conclusion

The usability evaluation revealed very vital comments and usability flaws that were essential for the eMCH tool to be upgraded. All evaluation metrics showed statistically significant performance differences among the three groups of users. The eMCH tool was found to be feasible and acceptable as reported by end-users. Therefore, the errors and usability flaws of the eMCH tool should be fixed before deployment to other healthcare settings and before considering for scale-up.

## Supporting information

**S1 Fig. Main interfaces of the eMCH application.**
(PNG)

**S2 Fig. ePartograph interface used to enter parameters used to draw partograph.**
(PNG)

**S3 Fig. WHO safe child-birth checklists interface.**
(PNG)

## Acknowledgments

This research team acknowledges Bahir Dar Institute of Technology for providing library and internet facilities while we were conducting this research. We are also very grateful to Bahir Dar University College of Medicine and Health Sciences for providing ethical clearance.

## Author Contributions

**Conceptualization:** Esubalew Alemneh, Tegegn Kebebaw, Dabere Nigatu, Muluken Azage, Eyaya Misgan, Enyew Abate.

**Data curation:** Esubalew Alemneh, Dabere Nigatu.

**Formal analysis:** Esubalew Alemneh, Dabere Nigatu.

**Funding acquisition:** Esubalew Alemneh, Tegegn Kebebaw, Muluken Azage.

**Investigation:** Esubalew Alemneh, Tegegn Kebebaw, Dabere Nigatu, Muluken Azage, Eyaya Misgan, Enyew Abate.

**Methodology:** Dabere Nigatu, Muluken Azage.

**Project administration:** Esubalew Alemneh, Tegegn Kebebaw, Muluken Azage, Eyaya Misgan, Enyew Abate.

**Software:** Tegegn Kebebaw.

**Supervision:** Esubalew Alemneh.

**Validation:** Eyaya Misgan, Enyew Abate.

**Writing – original draft:** Dabere Nigatu, Muluken Azage.

**Writing – review & editing:** Esubalew Alemneh, Tegegn Kebebaw, Dabere Nigatu, Muluken Azage, Eyaya Misgan, Enyew Abate.

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
