## [Decision Letter · Decision Letter 0]

22 Dec 2023

PDIG-D-23-00370

E-maternal and child health application usability, feasibility and acceptability among healthcare providers in Amhara Region, Ethiopia

PLOS Digital Health

Dear Dr. Nigatu,

Thank you for submitting your manuscript to PLOS Digital Health. After careful consideration, we feel that it has merit but does not fully meet PLOS Digital Health's publication criteria as it currently stands. Therefore, we invite you to submit a revised version of the manuscript that addresses the points raised during the review process.

Please submit your revised manuscript within 60 days Feb 20 2024 11:59PM. If you will need more time than this to complete your revisions, please reply to this message or contact the journal office at digitalhealth@plos.org. Please include the following items when submitting your revised manuscript:

We look forward to receiving your revised manuscript.

Kind regards,

Haleh Ayatollahi

Section Editor

PLOS Digital Health

Journal Requirements:

Additional Editor Comments (if provided):

Reviewers' comments:

Reviewer's Responses to Questions

**Comments to the Author**

1. Does this manuscript meet PLOS Digital Health’s publication criteria? Is the manuscript technically sound, and do the data support the conclusions? The manuscript must describe methodologically and ethically rigorous research with conclusions that are appropriately drawn based on the data presented.

Reviewer #1: Yes

Reviewer #2: Yes

2. Has the statistical analysis been performed appropriately and rigorously?

Reviewer #1: Yes

Reviewer #2: Yes

3. Have the authors made all data underlying the findings in their manuscript fully available (please refer to the Data Availability Statement at the start of the manuscript PDF file)?

Reviewer #1: Yes

Reviewer #2: Yes

4. Is the manuscript presented in an intelligible fashion and written in standard English?

Reviewer #1: No

Reviewer #2: No

5. Review Comments to the Author

Reviewer #1: The description of the study and tool is not clear. Is this a tool for medical providers or for patients or both? What does it do? what paper system does it replace? Was the paper tool validated before the development of the electronic tool? how many providers and institutions used it before? 

Is there a central database where the data collected in the tool is stored for later review or is the data stored on the device in use? Are these mobile devices or desktop computers? What operating system is being used? 

There are multiple spelling and grammatical errors. In the abstract it says 'thick-aloud' in at least two different places.

The last sentence of the abstract is improper English: "Therefore, the tool should accommodate the recommended usability defects before deploying it in other healthcare settings." The correct term is not accommodate, nor defect. 

A more appropriate sentence would be: The errors and flaws of the tool should be fixed before deployment to other healthcare settings. 

The word 'tenet' is used out of context. Not sure what the authors meant in the sentence. The word is improperly used. 

The acronyms: ANC and PNC are used but not defined. 

How was the data de-identified for analysis? Is it possible to roll back for further analysis? 

Is there correlation between the age of the providers, their individual computer literacy and the results of the study regardless of the education level? Did older and less computer literate subjects make more errors? 

When the system has to be 'restarted' do they have to go back to the beginning and start over again or does the system save the data that is correct and the user can resume data entry where the error occurred? 

Aside from data entry, is the tool and its contents used afterwards? or is it used only for tracking on the spot? 

Figures:

Figure 1: what do the red question marks mean? 

The delivery information is on top of the graph and difficult to read.

Figure 2: the word efficiency is cut out

Figure 3: what is the difference between guidance and assistance? The figure does not stand on its own. 

I recommend correcting the spelling and grammatical errors.

Add a section with a brief description of the tool so the reader can better understand what the tool is about. 

Description of why is an electronic tool better than a paper tool. 

What is the return on investment of the electronic tool?

Compare age-computer literacy and usability results. The variables are there but there are no correlations between the variables. 

An English editor is necessary to make sure the correction of the grammatical errors reflects what the authors want to describe.

Reviewer #2: I would like to commend how the different parameters were explained in detail. However, some grammatical errors were seen in the Method and Materials section. It would also be better if the statistical analysis used was explained.

6. PLOS authors have the option to publish the peer review history of their article (what does this mean?). If published, this will include your full peer review and any attached files.

**Do you want your identity to be public for this peer review?** For information about this choice, including consent withdrawal, please see our Privacy Policy.

Reviewer #1: No

Reviewer #2: Yes: Arianne Justine Tan Obeles

---

## [Decision Letter · Decision Letter 1]

15 Mar 2024

PDIG-D-23-00370R1

Electronic maternal and child health application usability, feasibility and acceptability among healthcare providers in Amhara region, Ethiopia

PLOS Digital Health

Dear Dr. Nigatu,

Thank you for submitting your manuscript to PLOS Digital Health. After careful consideration, we feel that it has merit but does not fully meet PLOS Digital Health's publication criteria as it currently stands. Therefore, we invite you to submit a revised version of the manuscript that addresses the points raised during the review process.

Please submit your revised manuscript within 60 days May 14 2024 11:59PM. If you will need more time than this to complete your revisions, please reply to this message or contact the journal office at digitalhealth@plos.org. Please include the following items when submitting your revised manuscript:

We look forward to receiving your revised manuscript.

Kind regards,

Haleh Ayatollahi

Section Editor

PLOS Digital Health

Journal Requirements:

Additional Editor Comments (if provided):

Reviewers' comments:

Reviewer's Responses to Questions

**Comments to the Author**

1. If the authors have adequately addressed your comments raised in a previous round of review and you feel that this manuscript is now acceptable for publication, you may indicate that here to bypass the “Comments to the Author” section, enter your conflict of interest statement in the “Confidential to Editor” section, and submit your "Accept" recommendation.

Reviewer #1: All comments have been addressed

Reviewer #3: (No Response)

2. Does this manuscript meet PLOS Digital Health’s publication criteria? Is the manuscript technically sound, and do the data support the conclusions? The manuscript must describe methodologically and ethically rigorous research with conclusions that are appropriately drawn based on the data presented.

Reviewer #1: Yes

Reviewer #3: (No Response)

3. Has the statistical analysis been performed appropriately and rigorously?

Reviewer #1: Yes

Reviewer #3: (No Response)

4. Have the authors made all data underlying the findings in their manuscript fully available (please refer to the Data Availability Statement at the start of the manuscript PDF file)?

Reviewer #1: Yes

Reviewer #3: (No Response)

5. Is the manuscript presented in an intelligible fashion and written in standard English?

Reviewer #1: Yes

Reviewer #3: (No Response)

6. Review Comments to the Author

Reviewer #1: Thank you for addressing the comments and submitting the reviewed manuscript. It reads much better improving the readability. 

Very interesting tool. I hope the implementation goes smoothly. I can see the tool as a great asset for busy labor and delivery settings.

Reviewer #3: Dear Authors, I read your paper on eMCH adoption, usability, and feasibility among HCPs. The aim of the work is novel and pertinent to the journal's scope. As a reviewer, I've few questions for you and suggestions to enrich the paper further. I've also recommended to read and cite some relevant research papers to enrich the literature section.

Questions:

-----------

1. How do you define 'error' and 'restarts'?

2. Why have you used KW test and not One-way ANOVA to compare samples?

3. Please describe 'think aloud protocol'

4. How do you define 'integrative, interoperable and vendor-neutral digital health platform'?

Suggestions:

--------------

1. Please avoid 'we', 'our', 'us' in the text. 

2. Show few screenshots of the eMCH tool for generating more relevance.

Recommendations: 

Chattopadhyay, S. Shinha, P. “Understanding Factors Impacting COVID Vaccination in India: A Preliminary Report”. Quantum Journal of Medical and Health Sciences, (2021), 1(3): 18-31. 

Chattopadhyay S., Daneshgar F. – “An Awareness Net Collaborative Model for Schizophrenia Management”, International Journal of Advanced Intelligence Paradigms (2013); 5(3): 217-232.

Chattopadhyay S., – “A Prototype Depression Screening Tool for Rural Healthcare: A Step towards e-Health Informatics”, Journal of Medical Imaging and Health Informatics (2012); 2(3): 244-249

Chattopadhyay S., Sahu S.K., – “A Predictive Stressor-integrated Model of Suicide Right from One’s Birth: a Bayesian Approach”, Journal of Medical Imaging and Health Informatics (2012); 2(2):125-131

Chattopadhyay S., Saurabh R., Land L., Acharya U. R. – “Studying Infant Mortality Rate: A Data Mining Approach”, Health and Technology (2011); 1(1): 25-34

Daneshgar F., Chattopadhyay S. –“A Framework for Crisis Management in Developing Countries”. Intelligent Decision Technologies: an international journal (2011); 5(2): pp. 189-199. DOI: 10.3233/IDT-2011-0106

Chattopadhyay S. –"A Framework for Studying Perceptions of Rural Healthcare Staff and Basic ICT Support for e-health Use: An Indian Experience". Telemedicine and e-Health (2010); 16(1): pp. 80-88.

Ying W., Wimalasiri J., Ray P., Chattopadhyay S., Wilson C. - "An Ontology Driven Multi-Agent Approach to Integrated eHealth Systems". International Journal of e-Health and Medical Communications (2010); 1(1): pp. 28-40.

Li J, Land L.P.W, Ray P. Chattopadhyay S. –"E-Health Readiness Framework from Electronic Health Records Perspective". International Journal of Internet and Enterprise Management: Special Issue in Healthcare (2010); 6(4): pp. 326-348.

Ray P., Chattopadhyay S. – “Fuzzy Awareness Model for Disaster Situations”. Intelligent Decision Technologies: an international journal [Special Issue on Intelligent Decision Making in Dynamic Environments: Methods, Architectures and Applications] (2009); 3 (1): pp. 75-82.

Li J., Land L.P.W, Chattopadhyay S., Ray P. ‘E-Health Readiness Framework from Electronic Health Record Perspective’. Proceedings of SIG GlobDev Annual Workshop on ICT, Paris, France October 14-17 (2008)

Chattopadhyay S., Ray P., Land L. ‘Mining Demographic Health Indicators using Association-Correlations: A Technique for Improving the Healthcare Status’. Proceedings of 3rd International Conference on Software and Data Technology (ICSOFT 2008), Porto, Portugal July (2008), pp 315-320

Ganguly P, Chattopadhyay S., Ray P, Paramesh N. - 'An Ontology-based Framework for Managing Semantic Interoperability Issues in e-Health’. Proceedings of 10th IEEE Intl. Conf. on e-Health Networking, Applications and Service (HEALTHCOM 2008), pp. 73-78, Singapore (2008)

Li J, Land L, Chattopadhyay S., Ray P. – 'An Approach for E-health System Assessment & Specification'. In proceedings of 10th IEEE Intl. Conf. on e-Health Networking, Applications and Service (HEALTHCOM 2008), pp. 157-162, Singapore (2008).

Please note some of these research papers showcase early intervening models of eHealth and helped building current pillars of eHealth solutions/tools and required to be cited. 

Thank you for giving me this opportunity. Looking forward to read your revised version.

7. PLOS authors have the option to publish the peer review history of their article (what does this mean?). If published, this will include your full peer review and any attached files.

**Do you want your identity to be public for this peer review?** For information about this choice, including consent withdrawal, please see our Privacy Policy. 

Reviewer #1: Yes: Laritza M Rodriguez

Reviewer #3: Yes: Subhagata Chattopadhyay

---

## [Decision Letter · Decision Letter 2]

27 Mar 2024

Electronic maternal and child health application usability, feasibility and acceptability among healthcare providers in Amhara region, Ethiopia

PDIG-D-23-00370R2

Dear Mr. Nigatu,

We are pleased to inform you that your manuscript 'Electronic maternal and child health application usability, feasibility and acceptability among healthcare providers in Amhara region, Ethiopia' has been provisionally accepted for publication in PLOS Digital Health.

Best regards,

Haleh Ayatollahi

Section Editor

PLOS Digital Health

Reviewer Comments (if any, and for reference):

Reviewer's Responses to Questions

**Comments to the Author**

1. If the authors have adequately addressed your comments raised in a previous round of review and you feel that this manuscript is now acceptable for publication, you may indicate that here to bypass the “Comments to the Author” section, enter your conflict of interest statement in the “Confidential to Editor” section, and submit your "Accept" recommendation.

Reviewer #3: All comments have been addressed

2. Does this manuscript meet PLOS Digital Health’s publication criteria? Is the manuscript technically sound, and do the data support the conclusions? The manuscript must describe methodologically and ethically rigorous research with conclusions that are appropriately drawn based on the data presented.

Reviewer #3: Yes

3. Has the statistical analysis been performed appropriately and rigorously?

Reviewer #3: Yes

4. Have the authors made all data underlying the findings in their manuscript fully available (please refer to the Data Availability Statement at the start of the manuscript PDF file)?

Reviewer #3: No

5. Is the manuscript presented in an intelligible fashion and written in standard English?

Reviewer #3: Yes

6. Review Comments to the Author

Reviewer #3: The paper looks good to me.

7. PLOS authors have the option to publish the peer review history of their article (what does this mean?). If published, this will include your full peer review and any attached files.

**Do you want your identity to be public for this peer review?** For information about this choice, including consent withdrawal, please see our Privacy Policy.

Reviewer #3: **Yes: **Subhagata Chattopadhyay
